# Flexible Supercapacitors Based on Graphene/Boron Nitride Nanosheets Electrodes and PVA/PEIGel Electrolytes

**DOI:** 10.3390/ma14081955

**Published:** 2021-04-14

**Authors:** Chan Wang, Kuan Hu, Ying Liu, Ming-Rong Zhang, Zhiwei Wang, Zhou Li

**Affiliations:** 1CAS Center for Excellence in Nanoscience, Beijing Key Laboratory of Micro-Nano Energy and Sensor, Beijing Institute of Nanoenergy and Nanosystems, Chinese Academy of Sciences, Beijing 101400, China; wangchan@binn.cas.cn (C.W.); liuying@binn.cas.cn (Y.L.); wangzhiwei@binn.cas.cn (Z.W.); 2School of Nanoscience and Technology, University of Chinese Academy of Sciences, Beijing 101400, China; 3Department of Advanced Nuclear Medicine Sciences, The National Institute of Radiological Sciences, The National Institutes for Quantum and Radiological Science and Technology, Chiba 263-8555, Japan; kuan.hu@qst.go.jp (K.H.); zhang.ming-rong@qst.go.jp (M.-R.Z.)

**Keywords:** hydrogel electrolyte, flexible supercapacitor, graphene, boron nitride nanosheets

## Abstract

All-solid-state supercapacitors have gained increasing attention as wearable energy storage devices, partially due to their flexible, safe, and lightweight natures. However, their electrochemical performances are largely hampered by the low flexibility and durability of current polyvinyl alcohol (PVA) based electrolytes. Herein, a novel polyvinyl alcohol-polyethyleneimine (PVA-PEI) based, conductive and elastic hydrogel was devised as an all-in-one electrolyte platform for wearable supercapacitor (WSC). For proof-of-concept, we assembled all-solid-state supercapacitors based on boron nitride nanosheets (BNNS) intercalated graphene electrodes and PVA-PEI based gel electrolyte. Furthermore, by varying the electrolyte ions, we observed synergistic effects between the hydrogel and the electrode materials when KOH was used as electrolyte ions, as the Graphene/BNNS@PVA-PEI-KOH WSCs exhibited a significantly improved areal capacitance of 0.35 F/cm^2^ and a smaller ESR of 6.02 ohm/cm^2^. Moreover, due to the high flexibility and durability of the PVA-PEI hydrogel electrolyte, the developed WSCs behave excellent flexibility and cycling stability under different bending states and after 5000 cycles. Therefore, the conductive, yet elastic, PVA-PEI hydrogel represents an attractive electrolyte platform for WSC, and the Graphene/BNNS@PVA-PEI-KOH WSCs shows broad potentials in powering wearable electronic devices.

## 1. Introduction

The vigorous development of wearable smart devices presents new requirements for energy supply units [1,2]. With the advantages of fast charge/discharge capability, large energy density and power density, supercapacitors in film forms are gaining increasing interest in the field of lightweight, portable energy management devices for wearable electronics [3,4,5]. Many efforts were made to realize wearable supercapacitors with robust mechanical properties and excellent electrochemical performances, to ensure operation in complex and dynamic natural environments [6,7,8,9,10].

All-solid-state electrolyte plays an important role in determining the performance of the supercapacitor [10,11,12,13,14]. Polyvinyl alcohol (PVA) based hydrogel electrolytes were extensively studied due to their environment-friendly, easily accessible, and inexpensive attributes [15,16]. However, they usually exhibited poor water retention ability, easy fatigue under large deformation, and poor conductivity, which severely limits the performance of PVA-based supercapacitors. Polyethyleneimine (PEI) is a kind of cationic polymer, and the combination of PVA with PEI might be a possible strategy to solve the above shortcomings of PVA. As demonstrated in our previous work, the PVA-PEI hydrogels are ultra-stretchable and mechanically robust [17]. We further demonstrated that these hydrogels are excellent materials for piezoresistive sensors. However, whether the PVA-PEI hydrogels could be used as electrolytes of supercapacitors remains unknown. More importantly, how to integrate the flexibility of PVA-PEI hydrogels in fabricating “ideal” wearable SCs remains a formidable challenge.

In addition to electrolytes [18,19,20,21], the electrode materials and the interface between the electrode and the gel electrolyte are other important factors in deciding the overall performance of the devices. Graphene is a commonly used electrode material due to its large theoretical specific surface area (about 2600 m^2^/g) and ideal double-layer capacitance performance [22]. However, the output of graphene electrodes could be greatly affected by the intimate layer-to-layer stacking of the single graphene sheets, which prevents the formation of favorable three-dimensional structures important for the electrolyte ions’ storage and release [23]. Numerous methods, including chemical vapor deposition (CVD) [24], mechanical stripping and oxidation-reduction method [23], were reported for fabricating graphene-based supercapacitors. However, the complicated operation procedures of these methods make them poorly tolerable for large-scale preparation and application [25]. In this context, other simple and robust methods for fabricating ultrahigh performance graphene-based electrodes are urgently needed.

Boron nitride nanosheet (BNNS) is a two-dimensional hexagonal crystal material composed of carbon and nitrogen atoms in sp^2^-bonded [26]. With a similar crystal structure to graphene, BNNS could be easily assembled into graphene by a simple ultrasonic blending approach. Meanwhile, the integration of BNNS into graphene could stabilize and promote the capacitance of the graphene electrodes [27]. Herein, we proposed a wearable supercapacitor (WSC) based on graphene/BNNS electrodes and PVA/PEI-based hydrogel electrolytes (Figure 1). We aim to explore how BNNS affects the performance of the graphene electrodes; moreover, whether PVA-PEI is an “ideal” hydrogel electrolyte for all-solid-state supercapacitors. Last but not least, whether the graphene/BNNS electrodes and PVA-PEI electrolytes are a good combination for wearable SC devices.

## 2. Materials and Methods

### 2.1. Electrode Preparation

The carbon cloth was purchased from Guangzhou LiGe Technology Co., Ltd. and was used as the collector and scaffold for active materials. The dispersion of aminated carbon nanotubes (CNTs, CAS:1333-86-4, XFNANO, Nanjing, China) was prepared by mixing 0.8 g CNTs and 0.4 g sodium dodecylbenzene sulfonate (SDBS, CAS: 25155-30-0, Aladdin, Shanghai, China) in DI water (20 mL) with ultrasonication (80 W, 50 Hz) for 30 min. The CNTs were inserted into a carbon cloth (CC/CNTs) by vacuum filtration. 0.08 mg graphene (C, CAS: 7440-44-0, XFNANO, Nanjing, China), 0.01 g black carbon (CAS: 7440-44-0, XFNANO, Nanjing, China) and 0.008 g boron nitride nanosheets (BNNS, CAS: 10043-11-5, XFNANO, Nanjing, China) were mixed in 15 mL DI water with ultrasonication (80 W, 50 Hz) for 30 min. Then the G/BNNS was mounted on the CC/CNTs through vacuum filtration. The graphene electrode was prepared in the same way but without BNNS.

### 2.2. Electrolyte Preparation

The hydrogel (H) based on polyvinyl alcohol (PVA, CAS: 9002-89-5, 1799, Aladdin) and polyethyleneimine (PEI, CAS: 9002-98-6, MW = 600, Aladdin, Shanghai, China) was synthesized and used as an electrolyte with/without additional H_2_SO_4_ (CAS: 7664-93-9, Aladdin, Shanghai, China) or KOH (CAS:1310-58-3, Aladdin, Shanghai, China). For the electrolyte of H/KOH, the homogeneous solution was prepared by dissolving 2 g PVA, 1 g PEI and 0.56 g KOH (1 M) in 10 mL DI water and then heating in a water bath (90 °C) with magnetic stirring for two hours. Other electrolytes of PVA/H_2_SO_4_, PVA/KOH and H/H_2_SO_4_ were prepared using the same procedures.

### 2.3. Assembling the WSC

The WSC was assembled with two electrodes face to face and the electrolyte solution sandwiched between the two electrodes. The assembled WSC was then stored at −20 °C overnight. After thawing at room temperature for 12 h, the WSC can be used for further study.

### 2.4. Material Characterizations

The Tensile test of the hydrogel was tested by an ESM301/Mark-10 system (Mark-10 Corporation, New York, NY, USA) with a tensile speed of 50 mm·min^−1^. The samples for the Tensile test had a size of 20 × 10 × 1 mm^3^. Fourier transform infrared spectroscopy (FTIR) was obtained using a VERTEX80v spectrometer (Bruker, Karlsruhe, Germany). Scanning electron microscopy (SEM) and Energy Disperse Spectroscopy (EDS) images of the hydrogel were obtained using a Hitachi field emission scanning electron microscope (SU 8020, Hitachi, Tokyo, Japan).

### 2.5. Electrochemical Measurements

The cyclic voltammetry (CV), galvanic charge/discharge (GCD), electrochemical impedance spectroscopy (EIS) tests of the WSCs were performed with an electrochemical workstation (CHI 650E, Chenhua, Shanghai, China). The EIS was performed with an AC amplitude of 10 mV at an open circuit at a range of 100 kHz to 0.01 Hz.

The area capacitance was calculated by:$$C_{S}=\frac{1}{S\Delta V\nu }\int_{V_{1}}^{V_{2}}\mathrm{idV}$$
where ΔV is the cut-off voltage (i.e., 0.8 V), υ is the scan rate, S is the area of the tested device.

## 3. Results and Discussion

### 3.1. Properties Characterization of the PVA-PEI Based Hydrogel

Through a simple water bath reaction associated with a freezing/thawing method [17], the hydrogel elastomer with good transparency was synthesized (Figure 2a). The long-chain PVA and short-chain PEI formed a crosslinked network due to multiple physical and chemical intermolecular or intramolecular interactions, such as electrostatic attraction, segment entanglement and hydrogen bonding. The hydroxyl groups (−OH) of PVA and the amino groups (−NH_2_) of PEI may especially form tough hydrogen bonds, which provide quite a suitable environment for retaining abundant water molecules around the polymer chains. Moreover, the strong nucleophilic abilities of −NH_2_ and −NH- in PEI may hydrolyze water molecules, resulting in increased conductivity of the hydrogel. Compared to conventional PVA hydrogel electrolytes, the PVA-PEI system affords a more preferable three-dimensional framework for ions shuttling and storing, particularly for alkali ions.

The PVA-PEI hydrogel exhibited superior mechanical elasticity (Figure 2b). After freeze-drying treatment, the SEM images revealed that the hydrogel possesses micro/nano-structured pores or crackles, which may offset the gel volume variations during the charge/discharge processes. The FTIR spectrum was performed to verify the components of the hydrogels (Figure 2d). The broad peak ranging from about 3208 cm^−1^ to 3362 cm^−1^, associated with a weak peak around 1097 cm^−1^, were assigned to the ν-OH stretching of PVA. Similarly, the −NH− stretching of PEI could be found as a weak peak at 1042 cm^−1^ [17].

To study the stretchability of the hydrogel, we conducted a tensile stress–strain test. The PVA-PEI hydrogel exhibited excellent elastic properties, showing a breaking strength of up to 400 kPa and a breaking deformation of 425%, which are 2-fold and 1.5-fold improvements, respectively, compared to that of the PVA hydrogel with the same water content. The outstanding mechanical properties of the PVA-PEI hydrogel make them broad application prospects in wearable electronic devices.

### 3.2. Morphology and Electrochemical Performance of the Electrode

After obtaining the hydrogel electrolyte, we then assembled flexible electrodes. Among all kinds of electrode supporting materials, carbon clothes composed of conductive carbon fibers that are highly flexible, lightweight and have a large specific surface area, making them one of the most promising materials for WSCs. Figure 3a exhibited the surface structure of bared carbon cloth. After loading with CNTs (Figure 3b), the crisscrossing surface of the carbon cloth was fully covered by a dense layer of CNTs. Figure 3c,d shows the surface morphology of carbon cloth further loaded with graphene and BNNS. The graphene/BNNS composite has a layered structure, where the BNNS inserted into, or decorated on, the surface of graphene (Figure 3d). This demonstrates that the BNNS was well distributed and successfully prevented graphene from aggregation. This phenomenon was further confirmed by the element distribution map in EDS (Figure 3e).

### 3.3. Effect of BNNS on Electrochemical Properties of WSC

With the electrodes and hydrogel materials in hand, WSC was assembled (Figure 4a,b). It is worth mentioning that, the −OH of PVA and −NH_2_ of PEI could contribute to the tight binding of the hydrogel to the electrodes. Consequently, it helps to stabilize the sandwich structure during the repeated folding/unfolding process and improve performance stability during the charge/discharge cycle.

To study the influence of BNNS on the electrochemical performance of WSC, we designed two kinds of WSC devices, one is based on electrodes of graphene inserted with BNNS (denoted as G/BNNS@H) and the other one is based on electrodes with sole graphene (denoted as G@H) (More details could be found in Appendix A). The G/BNNS@H SC showed better electrochemical performance based on the comprehensive analysis of CV, GCD, rate curve and EIS test. All the CV curves of devices based on G@H and G/BNNS@H displayed shuttle-shaped closed loops over a potential window of 0 to 0.8 V at scan rates ranging from 1 mV/s to 100 mV/s. The CV curves of WSC based on G/BNNS@H performed more approximate right-angled curves than that of the device of G@H at the voltages of near 0 V or 0.8 V, indicating that BNNS doping leads to faster charge storage and release efficiencies (Figure 4c,e) [28]. At the same time, the triangle-shaped GCD curves of the two devices suggested superior capacitive performance and electrochemical reversibility (Figure 4d,f) [29]. The small voltage drop near 0.8 V is related to the internal resistance (∆R) of the supercapacitors, where the ∆R of G/BNNS@H is much less than that of G@H [30]. The areal capacitances at different scan rates were calculated and displayed in Figure 4g. G/BNNS@H showed an area capacitance of 0.0504 F/cm^2^, while that of G/H is 0.014 F/cm^2^. The electrochemical impedance (EIS) behavior of the WSCs was then studied (Figure 4h). G/BNNS@H shows a much smaller intercept with the x-axis than that of G@H (47.76 ohm/cm^2^ vs. 63.26 ohm/cm^2^), which means that G/BNNS@H has the lower equivalent series resistance (ESR) than that of G@H [31]. The smaller semicircle of G/BNNS@H in the middle frequency regions hints at a lower electron/ion charge-transfer resistance. In conclusion, the above characterizations show that BNNS effectively improves the electrochemical performance of graphene [32].

### 3.4. Enhanced Electrochemical Properties of WSC with Alkaline Hydrogel

Electrolyte ions play an important role in determining the conductance of the gel electrolyte and even the overall performance of the SCs. We next employed 1 M KOH or 1 M H_2_SO_4_ as electrolyte ions. The CV curves of G/BNNS@H/KOH presented a highly regular rectangle-like shape in a voltage window of 0 V to 0.8 V (Figure 5a). The corresponding GCD curves (Appendix A) showed triangle-like shapes. No significant voltage drop was observed when the direction of the current changes, implying a double-layer energy storage mechanism. In contrast, the CV (Figure 5b) curves of G/BNNS@H/H_2_SO_4_ revealed severe distortion, as so of the GCD curves (Appendix A), indicating that H_2_SO_4_ is a poor electrolyte choice for the PVA-PEI system.

Moreover, the electrochemical performance of WSCs with PVA-KOH/H_2_SO_4_ as electrolytes was tested (Appendix A). Among all four kinds of SCs, G/BNNS@H/KOH showed the largest areal capacitance of 0.35 F/cm^2^ at a scan rate of 1 mV/s (Figure 5c), while G/BNNS@H/H_2_SO_4_ is 0.037 F/cm^2^ at the same conditions. G/BNNS@PVA/H_2_SO_4_ and G/BNNS@PVA/KOH possessed a similar magnitude of areal capacitances. This indicates that the PEI is highly prone to acids. The significant enhancement of G/BNNS@H/KOH, with the addition of KOH, might because the alkaline electrolyte promoted the ionization of PEI and the stability of the hydrogel system.

Figure 5d showed the EIS plots of four WSCs at a frequency from 0.01 Hz to 100 kHz. Significant differences among the hydrogel electrolytes were observed. At high-frequency regions, ESRs were obtained from the intercept of the real axis. G/BNNS@H/KOH had the smallest impedance value of 6.03 ohm/cm^2^, followed by G/BNNS@H/H_2_SO_4_ and G/BNNS@PVA/H_2_SO_4_, their impedances are 7.23 and 7.46 ohm/cm^2^, respectively. G/BNNS@PVA/KOH showed the largest impedance of 19.4 ohm/cm^2^. These results indicated that the H/KOH has better contact with the electrode materials than the three other electrolytes. The Warburg region known as a straight line at 45° relative to axis ranging from the low frequency to the high frequency represents the ions diffusion impedance [33,34]. The slopes of H/KOH and H/H_2_SO_4_ are near 1, indicating excellent ion diffusion efficiencies owing to smaller contact resistance between the electrolyte and electrode [35]. The conductance calculated from ERS were displayed in Figure 5e. The G/BNNS@H/KOH showed the highest conductance and the G/BNNS@PVA/H_2_SO_4_ showed the lowest conductance.

Figure 5f shows the cyclic charge/discharge test of G/BNNS@H/KOH, G/BNNS@PVA/KOH and G@PVA/KOH at a current density of 0.53 mA/cm^2^. At the first 500 cycles, capacitances of all WSCs decreased fast. After 5000 cycles, the capacitance retention of G/BNNS@H/KOH was 81% of the original state, while G/BNNS@PVA/KOH and G@PVA/KOH were 75% and 71%, respectively.

### 3.5. The Flexibility and Application of WSC

Having demonstrated the optimal electrochemical performance of G/BNNS@H/KOH, the flexibility of G/BNNS@H/KOH was then explored. When the device was bent to 60°, 120°, and 180°, the CV curves at a scan rate of 4 mV/s were almost identical to the non-bending state, indicating excellent flexibility of the WSC (Figure 6a). As shown in Figure 6b, a cycling charge/discharge test at a current density of 0.53 mA/cm^2^ was carried out when the device was in its original state and a 60° bending state. The capacitance of WSC decreased rapidly during the initial 1000 cycles and gradually leveled off. After 5000 cycles of charging and discharging, the capacitance retention of the device was about 80%.

Benefitting from the thin film structure, the WSC could be connected in series or parallel to easily power different electronic devices and be integrated into clothing. When two or three devices were connected in series, the output voltage increases to 1.6 V and 2.4 V, which are two and three times to output voltage of a single device, respectively (Figure 6c). Correspondingly, the charging time extended to two or three times as long as one device. As a potential application, we tried to power a timer with three WSCs connected in series and achieved continuous operation for about one minute. Additionally, the WSCs were flexible enough to be anchored on the clothing and showed great potential in powering wearable electronic devices (Figure 6f).

## 4. Conclusions

In summary, a wearable all-solid-state supercapacitor with mechanical flexibility, outstanding electrochemical performance and superior stability was designed and fabricated based on graphene/BNNS electrode and hydrogel electrolyte. The highlights of this work originate following aspects. Firstly, a PVA-PEI based intrinsic conductance electrolyte was proposed and pioneeringly applied in WSC. Secondly, the introduction of BNNS not only stabilized the porous structures of graphene electrodes but also promoted the electrochemical performances of the WSCs, including folding and cycling stabilities. Thirdly, the conductive, yet elastic, PVA-PEI hydrogel represented an attractive electrolyte platform for WSC with alternative ions, like acids [4,7,8], alkali [3,28], neutral inorganic salts [6] and ionic liquids [36,37,38,39,40], et al. By varying the electrolyte ions, the electrochemical performances of WSCs were further enhanced. The WSC with graphene/BNNS@H/KOH achieved an areal capacitance of 0.35 F/cm^2^ and an ESR of 6.03 ohm/cm^2^. Moreover, the G/BNNS@H/KOH exhibited excellent flexibility and durability. After 5000 times charging and discharging test under repeated bending state, 81% capacitance was retained.

The all-solid-state supercapacitor with hydrogel electrolytes shows many advantages, such as safety, flexibility, lightweight, high energy density and power density. Distinguished from previously reported methods for fabricating the state-of-the-art WSC, our study here addressed the point from the systematic design rather than paid attention to either the electrolyte or the electrode materials. Through tailoring the electrolyte, electrode materials, and the interface chemistry between them, we successfully obtained a kind of WSC with satisfiable power density, flexibility, and durability. We envisage that the currently graphene/BNNS@H/KOH system will be further optimized by a more extensive screening and combination in the future.

## Figures and Tables

**Figure 1 materials-14-01955-f001:**
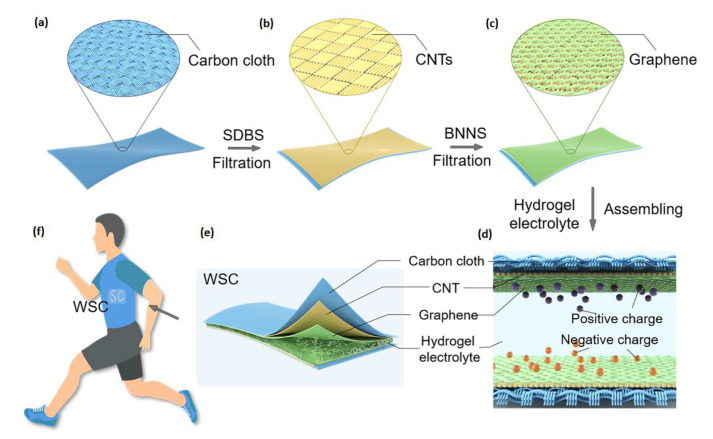
A schematic showing the fabrication process and structure of the wearable supercapacitor (WSC). (**a**) The carbon cloth was used as a current collector. (**b**) The carbon nanotubes (CNTs) were loaded on the carbon cloth through filtration, where sodium dodecylbenzene sulfonate (SDBS) was used as a dispersant. (**c**) The graphene was mounted on the CNTs layer through filtration, where boron nitride nanosheets (BNNS) was used to prevent graphene from collapsing. (**d**) The as-fabricated WSC was assembled with hydrogel used as a solid electrolyte. (**e**) The schematic structure of the WSC. (**f**) The WSCs could be integrated into clothing to power wearable devices.

**Figure 2 materials-14-01955-f002:**
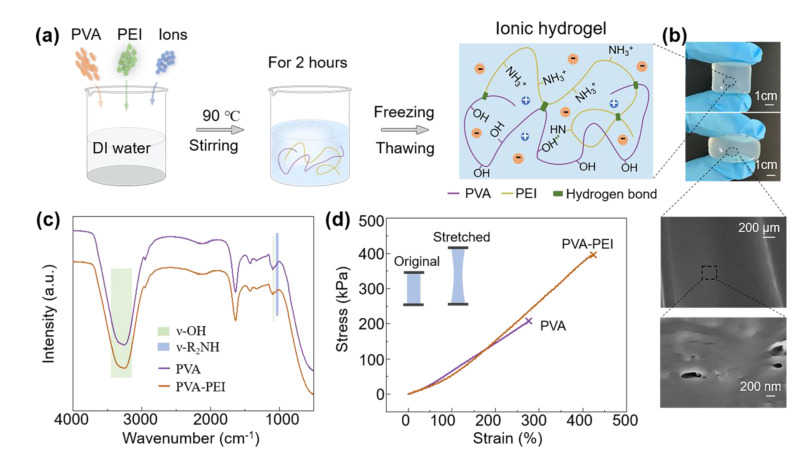
Schematic showing the fabrication process and the properties of the ionic hydrogel. (**a**) The preparation process and structure of the ionic hydrogel. (**b**) The as-fabricated hydrogel exhibited superior transparency and mechanical elasticity. The SEM images of the hydrogel showed a porous structure. (**c**) The Fourier transform infrared spectroscopy (FTIR) spectra of PVA-PEI-based ionic hydrogel. (**d**) The tensile stress–strain curves of PVA and PVA-PEI-based hydrogel.

**Figure 3 materials-14-01955-f003:**
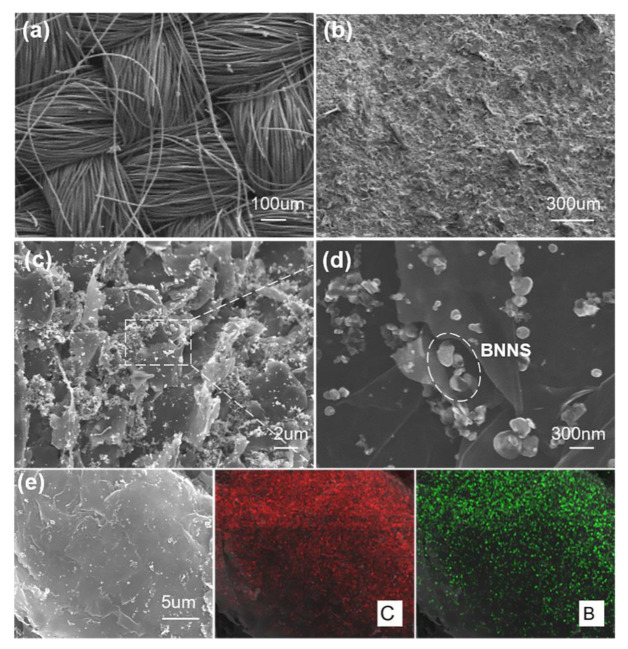
SEM images of the CC/CNT/graphene electrode. (**a**) SEM image of the carbon cloth. (**b**) SEM image of the carbon cloth loaded with CNTs. (**c**) SEM image of the carbon cloth loaded with CNT and graphene. (**d**) The BNNS inserting into layered graphene sheets to prevent collapse. (**e**) SEM images and the corresponding Energy Disperse Spectroscopy (EDS) elemental mapping of (**C**,**B**) for CC/CNT/graphene electrode.

**Figure 4 materials-14-01955-f004:**
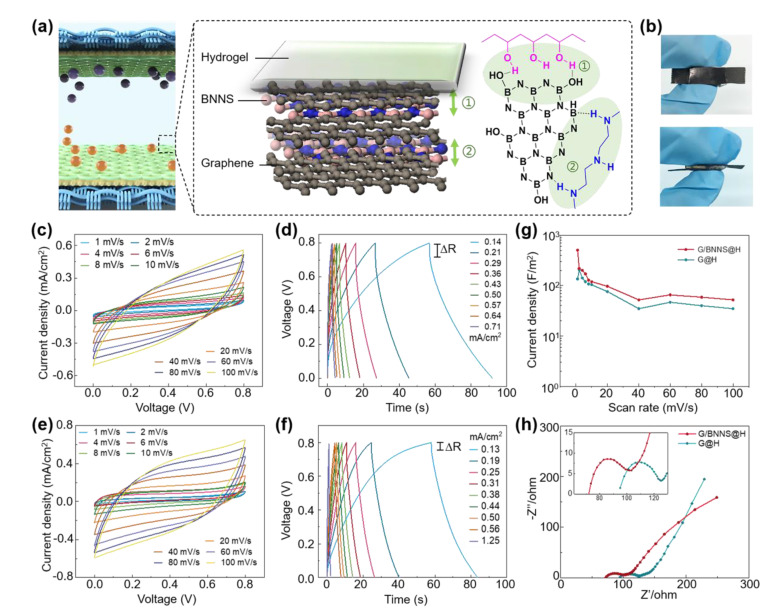
The electrochemical behavior of the WSCs. (**a**) The interaction among the electrolyte, BNNS and graphene. (**b**) the optical image of the WSC. CV (**c**) and galvanic charge/discharge (GCD) (**d**) curves of the WSC based on G@H. CV (**e**) and GCD (**f**) curves of the WSC based on G/BNNS@H. (**g**) Areal capacitances of WSCs at different scan rates. (**h**) Nyquist plot for WSCs based on G/BNNS@H and G@H.

**Figure 5 materials-14-01955-f005:**
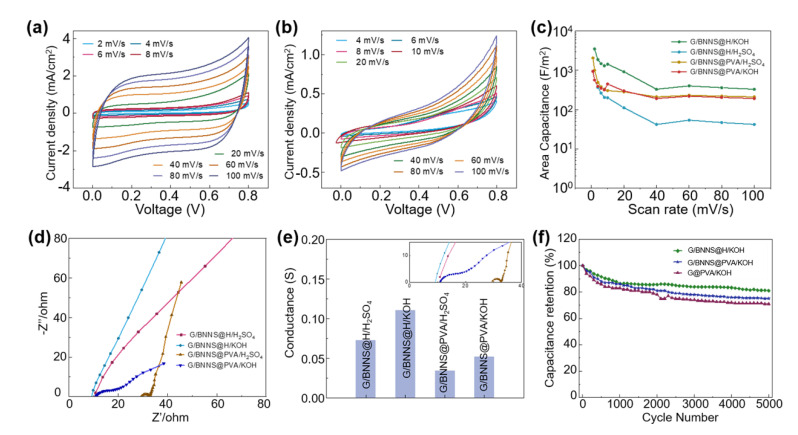
The electrochemical behavior of the WSCs. (**a**) CV curves of the WSC based on G/BNNS@H/KOH. (**b**) CV curves of the WSC based on G/BNNS@H/H_2_SO_4_. (**c**) Areal capacitance of WSCs at different scan rates. (**d**) Nyquist plot of WSCs. (**e**) The conductance of WSCs. (**f**) Cycling stability of WSCs at a current density of 0.53 mA/cm^2^.

**Figure 6 materials-14-01955-f006:**
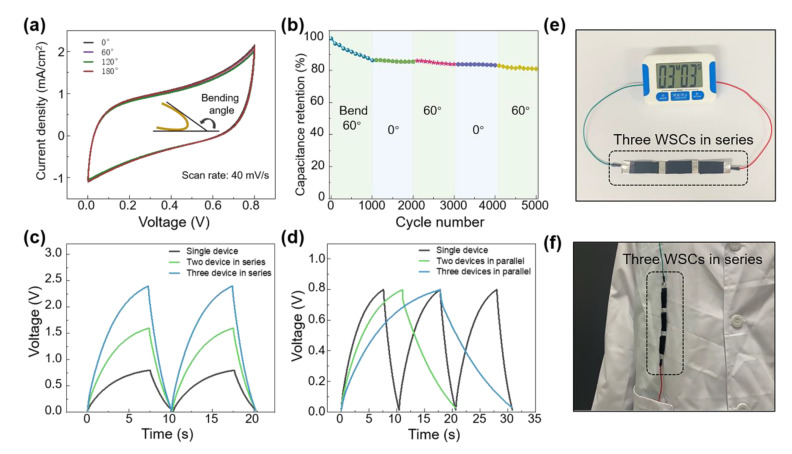
The electrochemical performance of the WSC with G/BNNS@H/KOH. (**a**) CV curves of the SC under different bending states obtained at a scan rate of 40 mV/s. (**b**) Cycling stability of WSCs with stage bending at a current density of 0.53 mA/cm^2^. (**c**) GCD curves of single/two/three WSCs connected in series. (**d**) GCD curves of single/two/three WSCs connected in parallel. (**e**) The timer powered by three WSCs linked in series. (**f**) The WSCs were anchored on cloth for powering the wearable electronic devices.

## Data Availability

All data is contained within the article.

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
