# Peer review of "Flexible Supercapacitors Based on Graphene/Boron Nitride Nanosheets Electrodes and PVA/PEI Gel Electrolytes"

_materials, 2021, doi:10.3390/ma14081955_

Round 1

Reviewer 1 Report

The paper “Flexible supercapacitors based on graphene/BNNS 2 electrodes and PVA/PEI gel electrolytes” by Wang et al. is focused on the use of a polymer (PVA-PEI)-based hydrogel as electrolyte in wearable supercapacitors. The results here obtained demonstrate the potential use of these polymer blends as electrolyte. Advances in the current electronic technology and in the development of electronic wearable devices is of large interest today, with many contributions to these fields emerging in the litterature. The topic is within the editorial line of nanomaterials. So, this journal is a good choice for this piece of research.

As far as this referee knows these results have not been published before. The topic cannot be considered as original, with many scientists working on similar issues, but the combined use of PVA and PEI results in devices with good deformation tolerance. In addition, the introduction of BNNS also results in electrochemical cycling stabilities.

The paper is clear and well-writen in my opinion. I have just detected a few typos (e.g. line 302, change “might be due to the that the alkaline electrolyte”… by might be due to the fact that the alkaline electrolyte… or ref 7. Change “Chem of Mater” by Chem. Mater.) that should be corrected.

The figures are also clear, although a larger font for the labels could be used to make them more readable at first sight.

The discussion is consistent with the experimental data. The discussion could be expanded in the section were figures 4, c, d, 3 and f are explained. The references the authors use are correct, but a few lines with a more explicit explanation of the meaning of the obtained results could improve the paper and the number of cites.

The conclusions are well-connected with the introduction, objectives of the paper and discussion. 

Author Response

Response to Reviewer 1 Comments

Dear editor:

We sincerely appreciate the editor and all reviewers for their thoughtful and professional critiques of our manuscript. We have adopted virtually all of the suggestions and revised the manuscript accordingly. We hope that the revised manuscript has been improved to the level of the editor and reviewers’ satisfaction. And we hope that you will now find it suitable for publication in Materials.

Our point-by-point responses to reviewers’ comments are detailed as following.

Reviewer: 1

The manuscript “Flexible supercapacitors based on graphene/BNNS electrodes and PVA/PEI gel electrolytes” deals with the production of a novel conductive and elastic hydrogel of PVA, to be used as a supercapacitor. Several analyses have been performed, obtaining good results. Therefore, the publication of this study is recommended; but after some revisions. In particular:

Response: Thank reviewer 1 for his/her supportive comments and valuable suggestions. These comments are very helpful to us. We have consulted relevant documents in order to provide the most satisfactory answers possibly. At the same time, certain adjustments were made to the article. We hope that the revised version will meet the publishing requirements.

  1. Please, define BNNS, PEI, SCs.

Response: Thank review 1 for the professional question. We are very sorry for making such a careless mistake. The BNNS, PEI and SCs were defined in abstract. The corresponding revisions read as:

Abstract: All-solid-state supercapacitors have gained increasing attention as wearable energy storage devices partially due to their flexible, safe, and lightweight nature. However, their electrochemical performances are largely hampered by the low flexibility and durability of current polyvinyl alcohol (PVA) based electrolytes. Herein, a novel polyvinyl alcohol- polyethyleneimine (PVA-PEI) based, conductive and elastic hydrogel was devised as an all-in-one electrolyte platform for wearable supercapacitor (WSC). For proof-of-concept, we assembled all-solid-state supercapacitors based on boron nitride nanosheets (BNNS) intercalated graphene electrodes and PVA-PEI based gel electrolytes. Furthermore, by varying the electrolyte ions, we observed synergistic effects between the hydrogel and the electrode materials when KOH was used as electrolyte ions, as the Graphene/BNNS@PVA-PEI-KOH WSCs exhibited a significantly improved areal capacitance of 3494.6 F/m2 and a smaller ESR of 9.04 ohm. Moreover, due to the high flexibility and durability of the PVA-PEI hydrogel electrolyte, the as-made WSCs behave excellent flexibility and cycling stability under different bending states and after 5000 times of cycles. Therefore, the conductive yet elastic PVA-PEI hydrogel represents an attractive electrolyte platform for WSC, and the Graphene/BNNS@PVA-PEI-KOH WSCs shows broad potentials in powering wearable electronic devices.

  1. Introduction. The state of the art about the preparation of aerogel-based supercapacitor can be enlarged. For this purpose, see, for example, the work of Sarno et al., SC-CO2-assisted process for a high energy density aerogel supercapacitor: The effect of GO loading, Nanotechnology, 2017, 28, Article number 204001; etc.

Response: Thank reviewer 1 for his/her careful reading and important comments. The article of Sarno et al. did a good work about SC-CO2-assisted process for a high energy density aerogel supercapacitor. After careful reading and studying, we added this article to the reference sequence and cited as Ref 5.

With the advantages of fast charge-discharge capability, large energy density and power density, supercapacitors in film forms are gaining increasing interest in the field of lightweight, portable energy management devices for wearable electronics [3-5].

  1. Results and discussion. Please, explain the influence of the morphology and porosity on the mechanical and electrochemical performance of the PVA samples produced.

Response:Thank reviewer 1 for his/her careful reading and professional comments. The PVA-PEI based hydrogel was designed to be 75% water content. Through Cryo-SEM, we observed porous structure (Figure 2b) which mainly due to the polymer skeleton left after the water volatilizes in the system.

The porous structure may help enhance the tensile and compressive capabilities of PVA-PEI based hydrogel. When the porous hydrogel suffers from tensile force, the directional stretching of these micro-nano pores enable polymer afford big deformation and dissipate large energy at the same time. When the porous hydrogel subjected compressive stress, the micro-nano pores provide compression space.

In addition, the morphology and porosity of the PVA-PEI based ionic electrolyte plays a positive role in enhancing the electrochemical performance. One the one hand, the micro-nano pores provide conductive paths for free ions participating in conduction. On the other hand, the porous polymer skeleton structure acts as a spacer between two electrodes to prevent short circuits.

  1. Conclusions. Please, rewrite in a more critical way.

Response: Thank review 1 for the professional question. During past several days, we have made much effort to modify the conclusion in a more critical way. We wish the revised manuscript will satisfy reviewer’s requirement. The modified conclusion read as below:

In summary, a wearable all-solid-state supercapacitor with mechanical flexibility, outstanding electrochemical performance and superior stability was designed and fabricated based on graphene/BNNS electrode and hydrogel electrolyte. The highlights of this work originate following aspects. Firstly, a PVA-PEI based intrinsic conductance electrolyte was proposed and pioneeringly applied in WSC. Secondly, the introduction of BNNS not only stabilized the porous structures of graphene electrodes but also promoted the electrochemical performances of the WSCs, including folding and cycling stabilities. Thirdly, the conductive yet elastic PVA-PEI hydrogel represented an attractive electrolyte platform for WSC with alternative ions, like acids[4,7,8], alkali[3,28], metallic inorganic salts[6] and ionic liquids[36-40], el al.. By varying the electrolyte ions, the electrochemical performances of WSCs were further enhanced. The WSC with graphene/BNNS@H/KOH achieved an area capacitance of 3494.6 F/m2 and a ESR of 9.04 ohm. Moreover, the G/BNNS@H/KOH exhibited excellent flexibility and durability. After 5000 times charging and discharging test under repeated bending state, 81% capacitance was retained.

   All-solid-state supercapacitor with hydrogel electrolytes shows many advantages, such as safety, flexibility, lightweight, high energy density and power density. Distinguished from previously reported methods for fabricating the state-of-the-art WSC, our study here addressed the point from systematic design rather than paid attention to either the electrolyte or the electrode materials. Through tailoring the electrolyte, electrode materials, and the interface chemistry between them, we successfully obtained a kind of WSC with satisfiable power density, flexibility, and durability. We envisage that the currently graphene/BNNS@H/KOH system will be further optimized by a more extensively screening and combination in the future.

Reviewer 2 Report

The manuscript “Flexible supercapacitors based on graphene/BNNS electrodes and PVA/PEI gel electrolytes” deals with the production of a novel conductive and elastic hydrogel of PVA, to be used as a supercapacitor. Several analyses have been performed, obtaining good results. Therefore, the publication of this study is recommended; but after some revisions.

In particular:

- Abstract. Please, define BNNS, PEI, SCs.

- Introduction. The state of the art about the preparation of aerogel-based supercapacitor can be enlarged. For this purpose, see, for example, the work of Sarno et al., SC-CO2-assisted process for a high energy density aerogel supercapacitor: The effect of GO loading, Nanotechnology, 2017, 28, Article number 204001; etc..

- Results and discussion. Please, explain the influence of the morphology and porosity on the mechanical and electrochemical performance of the PVA samples produced.

- Conclusions. Please, rewrite in a more critical way.

Author Response

Response to Reviewer 2 Comments

Dear editor:

We sincerely appreciate the editor and all reviewers for their thoughtful and professional critiques of our manuscript. We have adopted virtually all of the suggestions and revised the manuscript accordingly. We hope that the revised manuscript has been improved to the level of the editor and reviewers’ satisfaction. And we hope that you will now find it suitable for publication in Materials.

Our point-by-point responses to reviewers’ comments are detailed as following.

Reviewer: 2

The paper “Flexible supercapacitors based on graphene/BNNS 2 electrodes and PVA/PEI gel electrolytes” by Wang et al. is focused on the use of a polymer (PVA-PEI)-based hydrogel as electrolyte in wearable supercapacitors. The results here obtained demonstrate the potential use of these polymer blends as electrolyte. Advances in the current electronic technology and in the development of electronic wearable devices is of large interest today, with many contributions to these fields emerging in the literature. The topic is within the editorial line of nanomaterials. So, this journal is a good choice for this piece of research.

As far as this referee knows these results have not been published before. The topic cannot be considered as original, with many scientists working on similar issues, but the combined use of PVA and PEI results in devices with good deformation tolerance. In addition, the introduction of BNNS also results in electrochemical cycling stabilities.

Response: Thank reviewer 2 for his/her supportive comments and valuable suggestions. For the past several days, we have made much effort to improve/modify the manuscript. We wish the revised manuscript will satisfy reviewer’s requirement.

  1. The paper is clear and well-writen in my opinion. I have just detected a few typos (e.g. line 302, change “might be due to the that the alkaline electrolyte”… by might be due to the fact that the alkaline electrolyte… or ref 7. Change “Chem of Mater” by Chem. Mater.) that should be corrected.

Response: Thank reviewer 2 for careful reading and useful suggestions. We are very sorry for making such a careless mistake. We have revised the error above mentioned. Furthermore, we have carefully checked and polished the main text of manuscript and supplementary materials in the past several days, and made corrections to some grammatical or vocabulary errors.

  1. The figures are also clear, although a larger font for the labels could be used to make them more readable at first sight.

Response: Thank reviewer 2 for careful reading and kindly reminder. The too small labels in figures did cause difficult reading. During the last several days, we reassembled and enlarged labels to ensure the legibility of every figure. The revised figures could be found in the manuscript.

  1. The discussion is consistent with the experimental data. The discussion could be expanded in the section were figures 4, c, d, e and f are explained. The references the authors use is correct, but a few lines with a more explicit explanation of the meaning of the obtained results could improve the paper and the number of cite.

Response: Thank reviewer 2 for careful reading and important suggestions. We are sorry for too concise description of Figure 4c-4e, which has caused readers' confusion. Then, we gave a more explicit explanation to Figure 4c-4e, and the revised text read as below:

All the CV curves of devices based on G@H and G/BNNS@H displayed shuttle-shaped closed loops over a potential window of 0 to 0.8 V at scan rates ranging from 1 mV/s to 100 mV/s. The CV curves of WSC based on G/BNNS@H performed more approximate right-angle curves than that of the device of G@H at the voltages of near 0 V or 0.8 V, indicating BNNS doping leads to faster charge storage and release efficiencies (Figure 4c and 4e) [28]. At the same time, the triangle-shaped GCD curves of the two devices suggested superior capacitive performance and electrochemical reversibility (Figure 4d and 4f) [29]. The small voltage drop near 0.8 V is related to the internal resistance (∆R) of the supercapacitors, where the ∆R of G/BNNS@H is much less than that of G@H [30].

  1. The conclusions are well-connected with the introduction, objectives of the paper and discussion. 

Response: Thanks for reviewer 2 careful reading and approval.

Reviewer 3 Report

The authors study a novel elastic and conductive hydrogel for wearable electronics. They study both the mechanical as well electrical properties, finding interesting results.

The work is generally interesting. The work is pertinent, in scope, to the Journal as well as well-written. I recommend publication once the following points are addressed carefully.

- Due to their superior dielectric properties, have the authors considered ionic liquid-based electrolytes? There are relevant works from the Kozinsky group (theory, DOI: 10.1021/acs.jpcb.0c01089, DOI: 10.1021/acs.nanolett.9b03705, DOI: 10.1016/j.jpowsour.2019.04.085, DOI: 10.1021/acs.jpclett.9b00798) and the Schönhoff group (experiment, DOI: 10.1002/aesr.202000078, DOI: 10.1021/acs.jpcb.9b11330, DOI: 10.1021/acs.jpcb.9b11051) that the authors could cite and include in the discussion.

- Are the reported stress-strain curves in the linear and elastic regime? Have the authors checked for hysteresis effects?

- Figure 4, Figure 5, and Figure 6 are too small for publication, all the text is hardly visible.

Author Response

Response to Reviewer 3 Comments

Dear editor:

We sincerely appreciate the editor and all reviewers for their thoughtful and professional critiques of our manuscript. We have adopted virtually all of the suggestions and revised the manuscript accordingly. We hope that the revised manuscript has been improved to the level of the editor and reviewers’ satisfaction. And we hope that you will now find it suitable for publication in Materials.

Our point-by-point responses to reviewers’ comments are detailed as following.

Reviewer: 3

The authors study a novel elastic and conductive hydrogel for wearable electronics. They study both the mechanical as well electrical properties, finding interesting results. The work is generally interesting. The work is pertinent, in scope, to the Journal as well as well-written. I recommend publication once the following points are addressed carefully.

Response: Thank reviewer 3 for his/her valuable time and professional suggestions. These comments are very helpful to us. In order to provide satisfactory answer as possible, we consulted the relevant documents. At the same time, certain adjustments have been made to the article. We hope that the revised edition will meet the publication requirements.

  1. Due to their superior dielectric properties, have the authors considered ionic liquid-based electrolytes? There are relevant works from the Kozinsky group (theory, DOI: 10.1021/acs.jpcb.0c01089, DOI: 10.1021/acs.nanolett.9b03705, DOI: 10.1016/j.jpowsour.2019.04.085, DOI: 10.1021/acs.jpclett.9b00798) and the Schönhoff group (experiment, DOI: 10.1002/aesr.202000078, DOI: 10.1021/acs.jpcb.9b11330, DOI: 10.1021/acs.jpcb.9b11051) that the authors could cite and include in the discussion.

Response: Thank reviewer 3 for careful reading and professional advices. In this work, we designed and fabricated a conductive yet elastic PVA-PEI hydrogel, which provides an attractive electrolyte for WSC. Furtherly, the electrochemical performance of WSC was optimized through doping hydrogel electrolyte with acids or alkalis. In fact, the PVA-PEI hydrogel provides a universal platform for other conductive carriers, such as neutral inorganic salts or ionic liquids. Since this work aims to introduce an intrinsic conductive gel-based electrolyte, no further research has been done about other ionic additives. Thanks for reviewer 3’s significant suggestion and recommended articles. We have listed part of these articles as cited references (Ref. 36-40) and further discussed other ionic additives in the conclusion part. The revised text and citations are as following:

In summary, a wearable all-solid-state supercapacitor with mechanical flexibility, outstanding electrochemical performance and superior stability was designed and fabricated based on graphene/BNNS electrode and hydrogel electrolyte. The highlights of this work originate following aspects. Firstly, a PVA-PEI based intrinsic conductance electrolyte was propsed and pioneeringly applied in WSC. Secondly, the introduction of BNNS not only stabilizes the porous structures of graphene electrodes but also promotes the electrochemical performances of the WSCs, including folding and cycling stabilities. Thirdly, the conductive yet elastic PVA-PEI hydrogel represents an attractive electrolyte platform for WSC with alternative ions, like acids[4,7,8], alkali[3,28], neutral inorganic salts [6] and ionic liquids[36-40], el al.. By varying the electrolyte ions, the electrochemical performances of WSCs were further enhanced. The WSC with graphene/BNNS@H/KOH achieved an area capacitance of 3494.6 F/m2 and a ESR of 9.04 ohm. Moreover, the G/BNNS@H/KOH exhibited excellent flexibility and durability. After 5000 times charging and discharging test under repeated bending state, 81% capacitance was retained.

  1. Are the reported stress-strain curves in the linear and elastic regime? Have the authors checked for hysteresis effects?

Response: Thank reviewer 3 for careful reading and significant comments. The stress-strain curves of PVA-PEI and PEI performed the approximate liner regime as shown in Figure R1a (as well as Figure 2d in manuscript). As for hysteresis effects, in our previous article (Small, 2020, 1904758), we have tested hysteresis curves of PVA-PEI hydrogel under 100%, 200%, 300% and 400% deformation (Figure R1b). The area enclosed by the loading and unloading

curves represent the energy dissipated in the cycle. Under 100% deformation, the PVA-PEI hydrogel behaved elastic deformation, achieved full recovery after unloading and dissipated little energy. With deformation increase, unrecoverable deformation occurs and the energy dissipated grows large. Considering that the maximum deformation of human joints or skin not exceed 65% (Sci. Adv. 2017, 3, e1602076), WSC could well meet the requirement in wearable applications.

Figure R1. (a) The tensile stress-strain curves of PVA and PVA-PEI hydrogel. (b) Hysteresis curves of PVA-PEI hydrogel with 100%, 200%, 300% and 400% deformation (Dates from Small, 2020, 1904758). (c) The dissipation energy of PVA-PEI hydrogel with 100%, 200%, 300% and 400% deformation.

  1. Figure 4, Figure 5, and Figure 6 are too small for publication, all the text is hardly visible.

Response: Thank reviewer 3 for careful reading and kindly reminder. The too small labels in figures did cause difficult reading. During the last several days, we reassembled and enlarged labels to ensure the legibility of every figure. The revised figures could be found in the manuscript.

Round 2

Reviewer 2 Report

The authors performed the modifications proposed by the Reviewer and improved the manuscript.

Author Response

point-by-point response

Dear editor and reviewer 2:

We sincerely appreciate the editor and reviewer 2 for their professional critiques and approval of our manuscript. On the version submitted before, we have double checked the possible errors in pictures and main text of the manuscript. Then, we revised several phrases to give a better description of our work. The modifications read as following:

Many efforts have been made to realize wearable supercapacitors with robust mechanical properties and excellent electrochemical performances, to ensure operation in complex and dynamic natural environment [6-10].

All-solid-state electrolyte plays an important role in determining the performance of supercapacitor [10-14].

Where ΔV is the cut-off voltage (i.e., 0.8 V), υ is the scan rate, S is the area of the tested device.

We wish the revision could meet editor and reviewer 2’s publication requirements of Materials.